# Elovl2-Ablation Leads to Mitochondrial Membrane Fatty Acid Remodeling and Reduced Efficiency in Mouse Liver Mitochondria

**DOI:** 10.3390/nu14030559

**Published:** 2022-01-27

**Authors:** Alexia Gómez Rodríguez, Emanuela Talamonti, Alba Naudi, Anastasia V. Kalinovich, Anna M. Pauter, Gustavo Barja, Tore Bengtsson, Anders Jacobsson, Reinald Pamplona, Irina G. Shabalina

**Affiliations:** 1Department of Molecular Biosciences, The Wenner-Gren Institute, Stockholm University, SE-10691 Stockholm, Sweden; alexiagomezrguez@gmail.com (A.G.R.); emanuela.talamonti@su.se (E.T.); anastasia.kalinovich@su.se (A.V.K.); annapauter@gmail.com (A.M.P.); tore.bengtsson@su.se (T.B.); anders.jacobsson@su.se (A.J.); 2Department of Animal Physiology II, Faculty of Biological Sciences, Complutense University, E-28040 Madrid, Spain; gbarja@bio.ucm.es; 3Department of Experimental Medicine, Biomedical Research Institute of Lleida (UdL-IRBLleida), University of Lleida, E-25198 Lleida, Spain; albanaudi@gmail.com (A.N.); reinald.pamplona@udl.cat (R.P.)

**Keywords:** docosahexaenoic acid (DHA) deficiency, mitochondrial function, polyunsaturated fatty acids, membrane permeabilization, oxidative damage markers, adenine nucleotide translocase

## Abstract

The fatty acid elongase elongation of very long-chain fatty acids protein 2 (ELOVL2) controls the elongation of polyunsaturated fatty acids (PUFA) producing precursors for omega-3, docosahexaenoic acid (DHA), and omega-6, docosapentaenoic acid (DPA*n*-6) in vivo. Expectedly, Elovl2-ablation drastically reduced the DHA and DPA*n*-6 in liver mitochondrial membranes. Unexpectedly, however, total PUFAs levels decreased further than could be explained by Elovl2 ablation. The lipid peroxidation process was not involved in PUFAs reduction since malondialdehyde-lysine (MDAL) and other oxidative stress biomarkers were not enhanced. The content of mitochondrial respiratory chain proteins remained unchanged. Still, membrane remodeling was associated with the high voltage-dependent anion channel (VDAC) and adenine nucleotide translocase 2 (ANT2), a possible reflection of the increased demand on phospholipid transport to the mitochondria. Mitochondrial function was impaired despite preserved content of the respiratory chain proteins and the absence of oxidative damage. Oligomycin-insensitive oxygen consumption increased, and coefficients of respiratory control were reduced by 50%. The mitochondria became very sensitive to fatty acid-induced uncoupling and permeabilization, where ANT2 is involved. Mitochondrial volume and number of peroxisomes increased as revealed by transmission electron microscopy. In conclusion, the results imply that endogenous DHA production is vital for the normal function of mouse liver mitochondria and could be relevant not only for mice but also for human metabolism.

## 1. Introduction

Dietary polyunsaturated fatty acids (PUFAs) have been shown to play important roles in human health, and mitochondria could be an essential component of PUFAs’ effects [1,2,3]. There is growing evidence that dietary docosahexaenoic acid (DHA) profoundly affects mitochondrial membrane phospholipid composition and mitochondrial function [1]. It has been previously shown that deficiency in other PUFAs (precursors for DHA) in the diet causes uncoupling of oxidative phosphorylation in rat liver mitochondria [4]. In addition, impaired levels of dietary 18:2*n*-6 lead to decreased activity of the enzyme cytochrome c oxidase and reduced mitochondrial respiration in rat hearts [5]. However, the role of endogenously synthesized PUFA vs. PUFA taken up from the diet is poorly investigated [6].

Several studies have identified the association between genetic variants in the *FADS2* and *ELOVL2* genes controlling endogenous PUFA synthesis and obesity-related conditions in adults and children [7,8,9]. However, the underlying mechanism connecting genetics and metabolism remains unknown. We have previously described the creation of *ELOVL2*-ablated (Elovl2 KO) mice and, in vivo, confirmed the fatty acid elongase ELOVL2 (ELOngation of Very Long-chain fatty acids protein 2) as a critical enzyme in the production of PUFAs with 24 carbon atoms [10]. Furthermore, Elovl2 KO mice display substantially decreased omega-3, DHA, and omega-6, docosapentaenoic acid (DPA*n*-6) in liver and serum [11]. Moreover, Elovl2 KO animals show metabolic changes (lower respiratory quotient) and resistance to diet-induced obesity [11], suggesting a potential role of mitochondria in regulating these metabolic processes.

It is known that membrane phospholipids control the function of mitochondrial proteins [12,13,14]. The degree of unsaturation of membrane fatty acids is correlated with reactive oxygen species (ROS) production in the mitochondrial respiratory chain [15,16], leading to lipid peroxidation and subsequent damage of cellular macromolecules and their dysfunction [17]. In addition, previous studies show that changes in phospholipid composition, e.g., impaired levels of the non-bilayer-forming lipid phosphatidylethanolamine [18] or cardiolipin [4,19], affect mitochondrial function by decreasing respiratory capacity. However, little is known about how changes in the levels of specific classes of endogenous PUFAs influence mitochondrial function [6]. Furthermore, it is essential to consider that mitochondrial fatty acid composition remodeling is partly independent of the endoplasmic reticulum where a significant amount of PUFAs are endogenously formed and Elovl2 is located [20,21,22]. Therefore, examining mitochondrial fatty acid composition and specifically DHA level will help unravel the origin of mitochondrial PUFA.

Thus, the present study aims to evaluate the effect of Elovl2 ablation on mitochondrial function and deduce the impaired metabolism in Elovl2 KO mice. We isolated liver mitochondria and analyzed their fatty acid composition, bioenergetics, and oxidative stress markers to address this question. As a result, we have found that the Elovl2 ablation in the endoplasmic reticulum drastically reduced the content of mitochondrial DHA (and other PUFAs) concomitantly with a reduction in mitochondrial efficiency despite the absence of oxidative damage and preserved content of respiratory chain proteins. The results highlight the importance of endogenous long-chain PUFAs production for proper mitochondrial function and mammalian metabolism. Furthermore, the findings presented here emphasize the importance of considering the interindividual genetic variability in endogenous long-chain PUFA production and is relevant for both dietetics and scholars working in molecular metabolism.

## 2. Materials and Methods

### 2.1. Animals

Twelve-week-old male Elovl2 KO mice were generated as described previously [10] and maintained on a 129SV/Sv strain backcrossed for at least ten generations. As a control, wild-type (WT) littermates of the same age were used. All animals were housed at 24 °C and maintained on a 12 h light:12 h dark cycle. Animals were fed ad libitum with standard chow (R70 Standard Diet, Lactamin, Stockholm, Sweden) and had free access to water. Under these standard conditions, no differences in phenotype (body weight, food intake, skin and fur appearance) were observed between WT and Elovl2 KO mice.

This study was approved by the Institutional Review Board on Animal Studies of Stockholm University and by the Animal Ethics Board of the North Stockholm region (protocol code 302/11 and date of 25 August 2014).

### 2.2. Liver Tissue Collection and Mitochondrial Isolation

Mice were anesthetized for 3 min with a mixture of 79% CO_2_ and 21% O_2_ and decapitated. One Elovl2 KO and one wild-type mouse were simultaneously processed each day. Livers were isolated immediately after decapitation, washed in ice-cold isolation buffer (210 mM mannitol, 70 mM sucrose, 20 mM TES, 1 mM EDTA, pH 7.35) to remove traces of blood, cut into small pieces, and homogenized with a Potter homogenizer with a Teflon pestle in 35 mL of the isolation buffer. Throughout the isolation process, tissues were kept at 0–2 °C. Mitochondria were prepared by differential centrifugation as described previously [23]. Liver homogenates were centrifuged at 8500× *g* for 10 min at 2 °C using a Beckman J2-21 M centrifuge. The resulting supernatant containing floating fat, peroxisomes and other small cellular organelles was discarded. The pellet was resuspended in 35 mL of ice-cold isolation buffer with modifications (added 0.2% (*w/v*) fatty-acid-free BSA) to remove fat contamination and centrifuged again at 800× *g* for 10 min to pellet cell debris and nuclei. The resulting supernatant was centrifuged at 8500*g* for 10 min. The final mitochondrial pellets were resuspended by homogenization in a small glass homogenizer in the same medium containing 0.2% BSA. The concentration of mitochondrial protein was measured using fluorescamine [24] with BSA as a standard.

Freshly isolated liver mitochondrial suspension was divided into three aliquots: 10 µL of mitochondrial suspension was supplemented with a protease inhibitor cocktail (CompleteTM Mini; Roche, Basel, Switzerland) (10:1) in an Eppendorf tube, placed in liquid nitrogen, and then stored at −80 °C until immunoblot analysis. Another 50 µL mitochondrial suspension was flushed with nitrogen gas to substitute the air in the Eppendorf tube to prevent oxidation and was used for chromatography. The majority of the freshly isolated mitochondria was kept in glass deeply immersed in ice and used for oxygen consumption measurement. 

### 2.3. Mitochondrial Fatty Acid Composition Analysis

Fatty acids from mitochondrial membranes were analyzed as methyl ester derivatives by gas chromatography (GC) as previously described [25]. Separation was performed by a DBWAX capillary column (30 m × 0.25 mm × 0.20 μm) in a GC System 7890A with a Series Injector 7683B and an FID detector (Agilent Technologies, Barcelona, Spain). Fatty acid methyl esters were identified by comparison with authentic standards (Larodan Fine Chemicals, Malmö, Sweden). Results are expressed as mol%. The following fatty acyl indices were also calculated: saturated fatty acids (SFA); unsaturated fatty acids (UFA); monounsaturated fatty acids (MUFA); polyunsaturated fatty acids (PUFA) from *n*-3 and *n*-6 series (PUFA*n*-3 and PUFA*n*-6); and average chain length (ACL) = [(Σ%Total14 × 14) + (Σ% Total16 × 16) + (Σ% Total18 × 18) + (Σ% Total 20 × 20) + (Σ% Total 22 × 22) + (Σ% Total 24 × 24)]/100. Finally, the density of double bonds in the membrane was calculated by the Double Bond Index, DBI = [(1 × Σmol% monoenoic) + (2 × Σmol% dienoic) + (3 × Σmol% trienoic) + (4 × Σmol% tetraenoic) + (5 × Σmol% pentaenoic) + (6 × Σmol% hexaenoic)].

### 2.4. Oxidation-Derived Protein Damage Markers

Glutamic SemiAldehyde (GSA), AminoAdipic SemiAldehyde (AASA), CarboxyEthyl-Lysine (CEL), CarboxyMethyl-Lysine (CML), and MalonDiAldehyde-Lysine (MDAL) were determined as trifluoroacetic acid methyl ester (TFAME) derivatives in acid-hydrolyzed, delipidated, and reduced mitochondrial protein samples by GC/MS [17] using an HP6890 Series II gas chromatograph (Agilent, Barcelona, Spain) with an MSD5973A Series detector and a 7683 Series automatic injector, an HP-5MS column (30-m × 0.25-mm × 0.25-µm), and the described temperature program [17]. Quantification was performed by internal and external standardization using standard curves constructed from mixtures of deuterated and non-deuterated standards. Analyses were carried out by selected ion-monitoring GC/MS (SIM-GC/MS). The ions used were: lysine and [2H8]lysine, *m*/*z* 180 and 187, respectively; 5-hydroxy-2-aminovaleric acid and [2H5]5-hydroxy-2-aminovaleric acid (stable derivatives of GSA), *m*/*z* 280 and 285, respectively; 6-hydroxy-2-aminocaproic acid and [2H4]6-hydroxy-2-aminocaproic acid (stable derivatives of AASA), *m*/*z* 294 and 298, respectively; CML and [2H4]CML, *m*/*z* 392 and 396, respectively; CEL and [2H4]CEL, *m*/*z* 379 and 383, respectively; and MDAL and [2H8]MDAL, *m*/*z* 474 and 482, respectively. The amounts of product were expressed as µmoles of GSA, AASA, CML, CEL, or MDAL per mol of lysine.

### 2.5. Mitochondrial Oxygen Consumption

Oxygen consumption rates were monitored with a Clark-type oxygen electrode (Yellow Springs Instrument) in a sealed chamber at 37 °C, as described previously [23] or using a high-resolution oxygraph (Oroboros Oxygraph-2K, Austria) as described previously [26]. Mitochondria were added in a concentration of 0.5 mg to 2.0 mL of a continuously stirred incubation medium (100 mM sucrose, 20 mM K-Tes, 50 mM KCl, 4 mM KH_2_PO_4_, 2 mM MgCl_2_, 1 mM EDTA, and 0.1% fatty-acid free BSA). Basal respiration of mitochondria was measured in the presence of 5 mM glutamate plus 3 mM malate. To measure oxidative phosphorylation, we added 750 µM ADP following 3 µg/mL oligomycin to inhibit phosphorylation and provide the respiratory control ratio. Finally, 0.8 µM of the chemical uncoupler FCCP (carbonyl cyanide 4-(trifluoromethoxy) phenylhydrazone) was added to check the maximal capacity of the electron transport chain. Oleate concentration-response traces were performed with the addition of oleate in indicated concentration in the presence of oligomycin. An inhibitor of adenine nucleotide translocase (ANT), carboxyatractyloside (CATR), was used to identify the involvement of the ANT in oleate-induced uncoupling.

### 2.6. Transmission Electron Microscopy

Transmission electron microscopy was used to visualize cellular ultrastructure. Liver slices were fixed in 2% glutaraldehyde and 0.5% paraformaldehyde in 0.1 M sodium cacodylate buffer containing 0.1 M sucrose (pH 7.4) at room temperature for 30 min and stored in a refrigerator. Fixed tissues slices were rinsed in 0.15 M sodium cacodylate buffer (pH 7.4), postfixed in 2% osmium tetroxide (pH 7.4) at 4 °C for 2 h, dehydrated in ethanol and acetone before being embedded in LX-112 (Ladd Research Industries, Burlington, VT, USA). Sections were contrasted with uranyl acetate followed by lead citrate and examined in a Tecnai 12 Spirit BioTWIN transmission electron microscope (FEI Company, Eindhoven, The Netherlands) at 100 kV. Digital images were taken using a Veleta camera (Olympus Soft Imaging Solutions, GmbH, Münster, Germany). For estimation of mitochondrial volume (Vv density of mitochondria), digital images at a final magnification of 8200× were randomly taken of the cytoplasm of liver cells from tissue. Printed digital images were used, and the volume density (Vv) of mitochondria was calculated by point counting using a 2 cm (cytoplasm) and 1.5 cm (mitochondria) square lattice, according to [27]. The number of peroxisomes was estimated per 1 µm^2^.

### 2.7. Immunoblotting

For immunoblot analysis, aliquots of freshly isolated mitochondrial suspensions were stored at −80 °C after supplementation with protease inhibitor cocktail (CompleteTM Mini; Roche). Protein concentrations of the thawed mitochondrial samples were re-quantified using the Lowry method. An equal volume of reducing sample buffer (0.5 M Tris-HCl, pH 6.8, 10% (wt/vol) SDS, 2.5% (vol/vol) glycerol, 100 mM dithiothreitol, and 0.5% (wt/vol) bromophenol blue) was added to each sample. Proteins were separated by SDS-PAGE in ordinary 12% and 10% polyacrylamide gel (12% for Cox4, voltage-dependent anion channel (VDAC) and adenine nucleotide translocase (ANT2) and 10% for OXPHOS). Gels were loaded with 20 μg of protein per sample. Equal protein loading was confirmed by ponceau red staining for the OXPHOS immunoblot membrane and with prohibitin protein as the loading control for the ANT2 immunoblot membrane. Proteins were transferred to polyvinylidene difluoride membranes (GE Healthcare Life Sciences) in 48 mM Tris-HCl, 39 mM glycine, 0.037 (wt/vol) SDS, and 15% (vol/vol) methanol using a semi-dry electrophoretic transfer cell (Bio-Rad Trans-Blot SD; Bio-Rad Laboratories, Hercules, CA, USA) at 1.2 mA/cm^2^ for 90 min. After transfer, the membrane was blocked in 5% milk in Tris-buffered saline-Tween for one hour at room temperature and probed with the indicated antibodies overnight at 4 °C. The immunoblot was visualized with appropriate horseradish peroxidase-conjugated secondary antibodies and enhanced chemiluminescence (ECL kit, GE Healthcare Life Sciences) in a charge-coupled device camera (Fuji Film, Tokyo, Japan). Antibodies used were as follows: a mixture of monoclonal antibodies against some of the structural components of oxidative phosphorylation (OXPHOS), including subunit NDUFB8 of complexes I, SDHB subunit of complex II and UQCRC2 subunit of complex III (Total OXPHOS Rodent Antibody cocktail, MS601; Mitosciences, OR, USA), dilution 1:10,000; Cox4 subunit of complex IV antibody (Santa Cruz Laboratories, sc-376731), dilution 1:2000; ATP5A subunit of complex V antibody (Thermo Fisher, Waltham, MA, USA, #459240) dilution 1:2000 and VDAC antibody (Cell Signaling Technology, Danvers, MA, USA, #4661S), dilution 1:1000; adenine nucleotide translocase (ANT2) antibody (Cell Signaling, #14671S), dilution 1:1000 and anti-Prohibitin antibody (abcam, ab75766), diluted 1:5000.

Quantification of western blot was performed by using a standard control, a mix of all the samples, which was loaded at least twice on each gel and used as a reference to calculate the amount of protein. 

### 2.8. Real Time qPCR

The total RNA was isolated with TRI Reagent (Sigma Aldrich, Saint Louis, MO, USA) following the manufacturer’s procedure. For real-time PCR, 500 ng of total RNA was reverse transcribed using random hexamer primers, deoxynucleoside triphosphates, MultiScribe reverse transcriptase, and RNase inhibitor (Applied Biosystems, Foster City, CA, USA). cDNA samples were diluted 1:10, and aliquots of 2 µL of the sample cDNA were mixed with SYBR Green Jump- Start Taq ReadyMix (Sigma Aldrich), pre-validated primers, and diethylpyrocarbonate-treated water and were measured in triplicate for each sample. Thermal cycling conditions were 2 min at 50 °C, 10 min at 95 °C, and 40 cycles of 15 s at 95 °C and 1 min at 60 °C, followed by melting curve analysis on a Bio-Rad CFX Connect Real-Time system. The ΔCT method (2^−ΔCT^) was used to calculate relative changes in mRNA abundance. Primers used were the Ant1 isoform (forward, 5′-GCCGGAAAGGGGCTGATATT-3′; reverse, 5′-AGAAAGCGTTGGCTCCTTCA-3′), the Ant2 isoform (forward, 5′-TGATGCAGTCTGGACGCAAA-3′; reverse,5′-GATCTTCCGCCAGCAGTCAA-3′) and 18S ribosomal RNA (forward, 5′-GGGCCTCGAAAGAGTCCTGTA-3′; reverse, 5′-TACCCACTCCCGACCCG-3′). Expression analysis was performed using the BioRad detection system.

### 2.9. Chemicals

The following chemicals were used for supplementation and additions: fatty acid-free bovine serum albumin (BSA), Fraction V (Cat#10775835001, Roche Diagnostics GmbH, Mannheim Germany); malate (Cat#M9138, Sigma); glutamate (Cat#G1626, Sigma); ADP (Cat##01905, Sigma). An inhibitor of adenine nucleotide translocase, carboxyatractyloside (CATR) (Cat#216200, Calbiochem, San Diego, CA, USA), was dissolved in 20 mM Tes, pH 7.2. Oleate (sodium salt) (Cat#O7501, Sigma) was dissolved in 50% ethanol; FCCP (Cat#C2920, Sigma) and oligomycin (Cat#O4876, Sigma) were dissolved in 95% ethanol. Used concentrations of ethanol do not affect mitochondrial function.

### 2.10. Statistics

Data were analyzed by Prism 4 software (GraphPad Software, San Diego, CA, USA) or KaleidaGraph version 5.0 by Synergy Software (Reading PA, USA). Differences between groups were analyzed by student’s t-test. A *p* value < 0.05 was considered significant.

## 3. Results

### 3.1. Enhanced Mitochondrial Volume and Number of Peroxisomes in Elovl2 KO Mice 

Previous studies of Elovl2 KO mice have shown that these mice are metabolically impaired [11,28], suggesting the possible involvement of mitochondrial mechanisms. Elovl2 is expressed in the liver, and the primary metabolic mechanism may occur in liver mitochondria. Therefore, an analysis of mitochondrial morphology and function became the goal of our study. 

Transmission electron microscopy of mouse hepatocytes in the fixed liver slices showed a large number of primarily round mitochondrial cross-sections, which occupied a large area of the cytosol (Figure 1A–D and Appendix A). 

Although most morphological features (shape and structure) of mitochondria were similar in wild-type and Elovl2 KO hepatocytes, mitochondrial volume was 21% higher in KO than in WT (Figure 1E). Thus, the higher mitochondrial volume could be one of the mechanisms of metabolic changes in Elovl2 KO mice.

The interesting finding by transmission electron microscopy was the increased number (by 44%) of peroxisomes in KO compared to WT animals (Figure 1F), likely as a compensatory mechanism for the absence of Elovl2 by enhancing the steps of DHA and DPA*n*-6 formation in the peroxisomes [29]. Notably, there were no visible differences in the ER, nucleus, or glycogen content between wild-type and Elovl2 KO livers (Figure 1A–D and Appendix A).

### 3.2. Elovl2 Ablation Leads to Pronounced PUFAs Deficiency in Mitochondrial Fatty Acid Composition

Earlier studies on Elovl2 KO mice identified that fatty acid composition in both phospholipids and triglycerides of liver tissue was significantly modified, showing almost complete absence of DHA and DPA*n*-6 [11]. Mitochondrial phospholipids contribute to total cellular phospholipids, and one can suggest that observed changes at the tissue level would be the same as at the mitochondrial level. It is known, however, that the origin of mitochondrial phospholipids is complex: some phospholipids are transferred as entire molecules from the endoplasmic reticulum, some are entirely built de novo within mitochondria, and some are modified there [20,21,22]. Therefore, the mitochondrial fatty acid composition could be very different from the tissues. Considering that Elovl2 activity occurs in the endoplasmic reticulum, we wonder how mitochondrial fatty acid composition is affected by Elovl2 ablation. Analysis of fatty acids from mitochondrial membranes as methyl ester derivatives by gas chromatography showed that the levels of omega-3, 22:6n-3, DHA, and omega-6, 22:5(*n*-6), DPA*n*-6 were significantly lower in Elovl2 KO mitochondrial extracts (DHA, 88% reduction and DPA*n*-6, 35% reduction) as compared with WT extracts (Figure 2A).

These results were expected and consistent with findings in whole tissue extracts [11], which directly reflects the absence of Elovl2 activity. Additionally, as expected, 22:5*n*-3 levels were significantly increased (doubled) in the Elovl2 KO mitochondria than in wild-type mitochondria (Figure 2A), a clear indication of the steps in which Elovl2 and peroxisomal oxidation are working in omega 3 (*n*-3) fatty acid remodeling [29].

However, effects of Elovl2 ablation on the omega 6 (*n*-6) pathway were less clear. There was no an increase in 22:4*n*-6 fatty acid (the closest precursor of DPA*n*-6) levels, and levels of several other precursors of *n*-6 fatty acids with various chain lengths (18:2*n*-6; 20:4*n*-6) were even lower in Elovl2 KO mitochondrial extract as compared with WT (Figure 2A). Such mitochondrial results contrast tissue results where high (not low) levels of these *n*-6 fatty acids have been observed, as expected [11].

A further exciting finding for mitochondria was the highest increase for fatty acid 18:1*n*-9 levels (38%), far from the Elovl2 remodeling point. These results highlight the relatively independent processes of mitochondrial fatty acid remodeling from tissue (endoplasmic reticulum) processes.

As a consequence of such mitochondria-specific fatty acid remodeling, the total levels of *n*-3 and *n*-6 PUFAs decreased by 35% and 16%, respectively, and the level of MUFA increased by 34% in mitochondria of Elovl2 KO as compared to WT (Figure 2B). In addition, the double bond index (DBI) (parameter directly reflecting fatty acid unsaturation degree) was 17% lower in the Elovl2 KO mice as compared to wild-type mice (Figure 2C). Furthermore, the acyl chain length (ACL) index was also significantly decreased in KO compared to WT (Appendix A).

Thus, Elovl2 ablation significantly changed the fatty acid composition in the mitochondrial membrane, and such changes reflect both the direct effect of the absence of Elovl2 activity and an additional unknown mechanism, affecting the PUFAs double bonds.

### 3.3. Absence of Oxidative Damage in Proteins of Elovl2 KO Mitochondria

Fatty acid unsaturation degree is related to oxidative stress and lipid peroxidation [15]. ROS attacks the double bonds of PUFA, leading to the production of reactive aldehydes (malondialdehyde (MDA) and 4-hydroxynonenal) and reduction in double bonds in polyunsaturated fatty acids [15]. Since DBI and PUFA levels were drastically low in Elovl2 KO (Figure 2B,C) and such reduction was not entirely due to the lack of Elovl2 activity, we suggested that high lipid peroxidation activity is responsible for this reduction. It is also known that the final lipid peroxidation products, aldehydes, are highly reactive molecules and ultimately react with surrounding molecules, mainly with proteins [17]. Therefore, high activity of lipid peroxidation (as a suggested mechanism of double bonds elimination) could be confirmed by the presence of high levels of oxidatively modified proteins in Elovl2 KO mitochondria.

To check our hypothesis, levels of oxidatively modified mitochondrial proteins were measured. However, in contrast to our prediction, the lipoxidation-dependent marker of protein modification, malondialdehyde lysine (MDAL), was significantly lower in Elovl2 KO mitochondria than in wild-type mitochondria; absolute values (micromoles/mole lysine) are shown in Figure 3, and the relative 28% reduction is shown in Appendix A. Thus, our hypothesis is rejected, and the mechanism behind the reduced PUFAs in Elovl2 KO mitochondria remains unclear.

Moreover, Elovl2 ablation entails membrane macromolecules’ resistance to other types of oxidative damage. The levels of the protein markers of glycoxidation, carboxyethyl lysine (CEL) and carboxymethyl lysine (CML), as well as the specific carbonyls, glutamic and aminoadipic semialdehyde (GSA and AASA), were not significantly different between wild-type and Elovl2 KO mice (Figure 3 and Appendix A).

Thus, due to fatty acid remodeling without ROS involvement, mitochondria in Elovl2 KO are PUFA deficient but protected from oxidative damage.

### 3.4. Reduced Mitochondrial Efficiency in Elovl2 KO Mice

The observed differences in mitochondrial volume and morphology between Elovl2 KO and WT mice (Figure 1), as well as severe DHA deficiency (and other PUFAs decrease) in membrane composition (Figure 2), predict the possible difference in mitochondrial function. Therefore, we analyzed the content of mitochondrial functional proteins and their oxidative phosphorylation activity in intact mitochondria.

Western blot analysis showed levels of inner membrane respiratory chain proteins (subunits of complexes I, II, III, and IV), phosphorylation protein ATP-synthase (subunit ATP5A), and outer membrane protein VDAC (Figure 4A). The inner mitochondrial membrane protein levels were not affected by Elovl2 ablation, whereas outer membrane protein VDAC was 32% higher in liver mitochondria isolated from Elovl2 KO mice than from WT mice (Figure 4B). Thus, Elovl2 KO mice are one of the unique mitochondrial models where the content and intactness (absence of oxidative damage) of respiratory chain proteins were preserved, despite the changes in membrane fatty acid composition.

We, therefore, next focused on analyzing the mitochondrial functions, namely the rates of mitochondrial oxygen consumption in the presence of glutamate + malate (typical liver mitochondrial substrate linked to respiratory complex I) and under different metabolic states. After addition of substrates, ADP was added to evaluate OXPHOS capacity, followed by the ATP synthase inhibitor oligomycin to evaluate mitochondrial membrane proton leak (LEAK flux), and finally the uncoupler FCCP to measure the maximal respiratory capacity through the electron transfer system (Figure 4C).

After substrate administration, the respiratory rate tended to be higher in liver mitochondria isolated from Elovl2 KO mice than from WT mice. Furthermore, respiratory rate became significantly higher in the presence of oligomycin (rate limited by the mitochondrial membrane proton leak and not by the respiratory chain) (Figure 4D). After ADP and FCCP administration, mitochondrial capacity tended to be lower in the Elovl2 KO mice (Figure 4D). Such changes in opposite directions indicated reduced mitochondrial efficiency and were revealed by calculating the respiratory control ratio (RCR) coefficients between ADP-stimulated and oligomycin-inhibited oxygen consumption rates and between FCCP-stimulated and oligomycin-inhibited oxygen consumption rates (Figure 4E). Mitochondrial RCR coefficients in Elovl2 KO mice were significantly lower (≈55%) than in WT mice (Figure 4E).

Given these results, we conclude that a decrease in long-chain PUFA content, especially DHA, in the mitochondrial membrane leads to decreased mitochondrial efficiency. Leaky (inefficient) mitochondria and increased total mitochondrial volume in liver tissue can be the mechanisms responsible for metabolic changes in Elovl2 KO mice.

### 3.5. Elovl2 KO Mitochondria Are More Sensitive to Fatty Acid-Induced Uncoupling

Having in hand an excellent mitochondrial model with increased basal proton leak and preserved respiratory chain proteins, we took the opportunity to study the mechanism of mitochondrial uncoupling in depth. We next looked at a mitochondrial inducible proton leak by natural inducers of proton leaks, fatty acids. 

The titration of mitochondria with oleate was performed in Elovl2 KO mitochondria in parallel with wild-type mitochondria (Figure 5A), and significant differences in concentration-response curves were observed (Figure 5A,B). In particular, the KO mitochondria were more sensitive to the uncoupling effect of oleate at concentration 60 µM oleate, where significantly higher oxygen consumption was observed in KO mitochondria than in WT mitochondria (Figure 5B).

Furthermore, although no difference between magnitudes of maximal rate of responses was observed in mitochondria of Elovl2 KO and WT mice (Figure 5B), the estimated EC50 (the dose induced 50% of response) was significantly higher in Elovl2 KO compared to WT (Figure 5C). Furthermore, notably, Elovl2 KO mitochondria are more easily inhibited (permeabilized) by a high concentration of oleate: 100 µM oleate induced profound inhibition of oxygen consumption (detergent-like effect) in Elovl2 KO mitochondria but not in WT mitochondria (Figure 5B). Thus, liver mitochondria deficient in PUFAs but with a preserved respiratory chain and without oxidatively damaged proteins have both high basal and inducible proton leak and easily undergo permeabilization.

### 3.6. Mitochondrial Adenine Nucleotide Translocase Is Involved in Fatty Acids-Induced Uncoupling in Elovl2 KO Mitochondria

To provide further insight into the mechanism of mitochondrial inefficiency, we investigated adenine nucleotide translocase (ANT), a membrane protein shown to be involved in mitochondrial uncoupling in the liver [30,31,32]. We used carboxyatractyloside (CATR), a specific and irreversible inhibitor of ANT [33,34] (Figure 6A). We found that a substantial fraction of the oleate-induced increase in oxygen consumption in liver mitochondria was inhibited by CATR (Figure 6A,B). This inhibitory effect appeared to be higher in KO mitochondria compared to WT (Figure 6C), suggesting that ANT mediates a significant part of the fatty acid-induced uncoupling effect in Elovl2 KO mitochondria.

While establishing the CATR effect, we noticed that prolonged exposure of mitochondria to oleate (of a notably small 60 µM concentration, below the concentration of 100 µM showing detergent-like effect in Figure 5A) and CATR induced a spontaneously developed change in oxygen consumption: the initial stable oxygen consumption slowly increased for 6–12 min and exhibited a short peak of oxygen consumption, followed by quickly developed inhibition (Figure 6A). Such a pattern of oxygen consumption changes is consistent with phenomena of spontaneous permeabilization. Notably, in Elovl2 KO mitochondria, the spontaneous permeabilization has developed much quicker than in WT mitochondria (Figure 6D).

Since the magnitude of the CATR effect may reflect the amount of ANT in mitochondria, we examined ANT protein levels in isolated mitochondria analyzed as described above. Western blot analysis revealed that the amount of ANT2 (the liver-specific isoform) was higher in Elovl2 KO mitochondria than in WT (Figure 6E,F). To confirm higher protein expression, Ant2 gene expression measurement in whole liver tissue was performed by Real-Time qPCR (Appendix A). Ant2 gene expression was 48% higher in the liver from Elovl2 KO than in wild-type liver. Higher Ant2 gene expression may reflect both higher ANT2 protein within the mitochondrial population as well as higher total mitochondrial biosynthesis (evident from higher tissue mitochondrial volume Figure 1E). Western blot and Real-Time qPCR of ANT1 were also performed, but no clear band was detected on western blot and very high Ct values on Real-Time qPCR (not shown), confirming that ANT2 is exclusively expressed in mouse liver.

Thus, fatty acid-induced uncoupled respiration in liver mitochondria is primarily mediated by ANT2. Furthermore, it seems that the contribution of this translocase in mitochondrial inefficiency is more significant in Elovl2 KO than in wild-type mitochondria. Furthermore, interestingly, the higher amount of ANT2 coincides with higher content of VDAC (Figure 4A,B), indicating the potential cooperation of these transport proteins.

## 4. Discussion

### 4.1. Possible Mechanisms of Impaired Mitochondrial Oxidative Phosphorylation in PUFA Deficient Mitochondria

The importance of lipid composition in the mitochondrial membrane for the electron transport chain and ATP synthesis has been suggested in various models [1,2,3,35]. Studied models in the liver are often pathologies (such as hepatic steatosis) and linked to oxidative stress (high production of ROS) [35,36]. PUFAs are considered to play a central role in these pathologies (as well as aging and age-related diseases) since within the mitochondrial membrane, PUFAs are the primary cellular target of ROS attack and contribute to the generation of oxidatively damaged proteins [16,17,37,38].

However, the Elovl2 KO mouse model studied here is an outstanding model. In contrast to the pathological mechanism described above of impairment of mitochondrial function via oxidatively modified mitochondrial PUFAs, the Elovl2 KO mitochondrial proteins are not damaged. Rather, the opposite was observed with the substantial decrease in the lipoxidation marker MDAL in the PUFA-deficient Elovl2 KO mice. Moreover, Elovl2 KO mice are resistant to hepatic steatosis induced by a high-fat diet [11]. Future studies of mitochondria in Elovl2 KO mice on a high-fat diet would help establish the possible link between resistance to oxidative damage and resistance to hepatic steatosis.

Regarding PUFA’s role in aging, it is worth mentioning here the discovery of the *ELOVL2* gene as the most extreme example of age-related DNA hypermethylation in human whole blood DNA [39]. Further studies are needed to understand whether *ELOVL2* hypermethylation only represents an indicator of chronological age or rather is functionally correlated with physiological status and specific clinical conditions. In this regard, one recent study showed that Elovl2 involves functional and anatomical age-associated changes in vivo, focusing on mouse retina, with direct relevance to age-related eye diseases [40].

Our study indicates that the deficiency in long-chain PUFAs (especially DHA), without modification of respiratory proteins, was sufficient to significantly reduce coefficients of respiratory control ratio (RCR), reflecting less coupled and less efficient oxidative phosphorylation in the Elovl2 KO mitochondria. This observation agrees with findings by [5,18] of lower oxidative phosphorylation in phospholipid remodeling mitochondria. Although no changes were seen in respiratory protein quantity, fatty acid remodeling of mitochondrial membranes can affect the formation of respiratory chain supercomplexes [41,42] and thus reduce oxidative phosphorylation efficiency. Another explanation for this inefficiency could be conformational changes that affect the catalytic centers of mitochondrial complexes and decrease their activity [19,43].

### 4.2. Mechanisms of Mitochondrial Inefficiency in Genetically Acquired PUFA-Deficiency Mice

Our findings suggest that mitochondria could play an essential role in the metabolic changes observed in the Elovl2 KO mouse model [11,28]. The high mitochondrial volume revealed by electron microscopy and, consequently, high total basal mitochondrial proton leak could contribute to high basal metabolism [44,45].

Furthermore, high oligomycin-insensitive respiration reflecting a basal proton leak at the mitochondrial level (independently of mitochondrial number) is another possible contributor to metabolic inefficiency of Elovl2 KO mice. There are ongoing debates in the literature about the mechanism of a basal proton leak in mitochondria [44,45,46,47,48]. Three alternatives are suggested: the first, that a proton leak is a pure mitochondrial membrane fatty acids/phospholipid characteristic; the second, that a proton leak is a pure mitochondrial membrane protein (such as UCP1) characteristic; and finally, that a proton leak occurs via membrane proteins, which are modified by surrounding phospholipids [47,48]. Aside from UCP1, adenine nucleotide translocase (ANT) is one of the membrane proteins that is involved in the mechanism of both basal and inducible mitochondrial proton leak [31,49,50,51]. Our earlier study [32] demonstrated that ANT1 is involved in a basal proton leak (in agreement with [52]), but that ANT2 is involved in fatty acid-induced uncoupling. ANT1 is not present in the liver; thus, the high basal proton leak (oligomycin-insensitive oxygen consumption) in liver mitochondria isolated from Elovl2 KO may be explained by the first alternative mechanism, where proton leak is a pure mitochondrial membrane fatty acids/phospholipid characteristic. Thus, our results suggested that PUFAs-deficient mitochondria are leaky.

Concerning the fatty acid-inducible mitochondrial uncoupling, several mechanisms of action have been discussed, including ANT involvement [53]. The content of ANT2 is high in isolated liver mitochondria of Elovl2 KO mice. Moreover, mitochondrial volume is also high, meaning the total enhanced level of ANT2 could significantly contribute to Elovl2 KO mouse liver metabolism. However, it needs to be considered that ANT-mediated uncoupling can only be observed under in vivo conditions where the level of endogenous non-esterified fatty acids is sufficiently high (reaching the level added in vitro). One such condition with high levels of non-esterified fatty acids due to hydrolysis of mitochondrial phospholipids was described for liver infection with *F.hepatica* [3]. Further research to identify additional pathologic or physiological in vivo conditions with elevated levels of non-esterified fatty acids is required to confirm the significance of ANT involvement in the mechanism of metabolic inefficiency.

### 4.3. Indications of Adaptive Structural Changes in Liver Mitochondria in Genetically Acquired PUFA-Deficiency Mice

In Elovl2 KO liver mitochondria, an upregulated level of the outer membrane protein VDAC was observed. VDAC is involved in numerous mitochondrial transport systems [22,54]. Our previous studies have shown outer membrane transport upregulation in mitochondria with respiratory chain deficiency [55]. In Elovl2 KO mice, the transport systems may respond to increased demand for lipid trafficking. Lipid exchange is essential for maintaining mitochondrial membranes, and it is driven at the endoplasmic reticulum-mitochondria juxtaposition [22,42,56]. Furthermore, endoplasmic reticulum-mitochondria contact sides change depending on food intake, i.e., increased under starvation but decreased in over-nutrition [21]. Such modification of endoplasmic reticulum-mitochondria contact sides could also happen in our model of PUFAs deficient mitochondria, which would be interesting to analyze to a deeper extent. Another interesting future perspective is related to the involvement of peroxisomal pathways and desaturases (Fads1 and Fads2) in adaptive modification of the fatty acid composition of mitochondrial membranes.

VDAC and ANT often cooperate as part of one transport complex through the inner and outer mitochondrial membrane [54]. Mitochondria with high levels of VDAC-ANT complexes are more prone to opening permeability transition pores and to apoptosis [57]. Observed patterns of spontaneous stimulation and following quick inhibition of oxygen consumption in Elovl2 KO liver mitochondria exposed to oleate and CATR may indicate easier ongoing permeabilization of the mitochondrial membranes. Such observation is consistent with [58] on CATR potentiation of spontaneous permeabilization in mitochondria with low membrane potential. The study of the detailed mechanism of such a phenomenon could be performed in future with more appropriate techniques.

### 4.4. The Physiological Value of Long Chain PUFAs as Nutrients

Using a genetically acquired PUFA-deficiency mouse model, we showed the significance of polyunsaturated fatty acids (especially long-chain, DHA) for proper mitochondrial function. Furthermore, our results helped to unravel the mechanism for dietary effects identified in numerous previous studies [1,2,5,59]. The omega-6 PUFAs, 18:2*n*-6 and 20:4*n*-6, and the omega-3 PUFA, 22:6*n*-3 (DHA), are the most abundant long-chain PUFAs in the phospholipids pool of the mitochondrial membrane (DHA content is more than 6 mol% of total mitochondrial lipids (Figure 2A)). Typical diets, however, contain relatively high amounts of 18:2*n*-6 and 20:4*n*-6 but are low in DHA content—especially the rodent chow diet, which lacks DHA typically. A study of Elovl2 KO mice has elucidated that the DHA within the liver mitochondrial membrane in chow-fed mice is endogenously derived via Elovl2 in the endoplasmic reticulum and peroxisomal fatty acid oxidation.

Thus, our study of a mouse model with genetically acquired fatty acid composition remodeling along with previous diet studies indisputably highlights the physiological value of PUFAs as nutrients. The knowledge presented here implies that endogenous long-chain PUFA production is vital for mice and human metabolism. Our findings are underlying several other studies on identifying genetic variants in the *ELOVL2* gene associated with obesity-related conditions [7,8,9]. Dietetics and scholars need to consider interindividual genetic variability in endogenous PUFA modification in humans coupled with PUFA derived from the diet when the relationship between PUFA intake and effects is investigated. These findings support the genetics accounting for interindividual variability in molecular metabolism.

## Figures and Tables

**Figure 1 nutrients-14-00559-f001:**
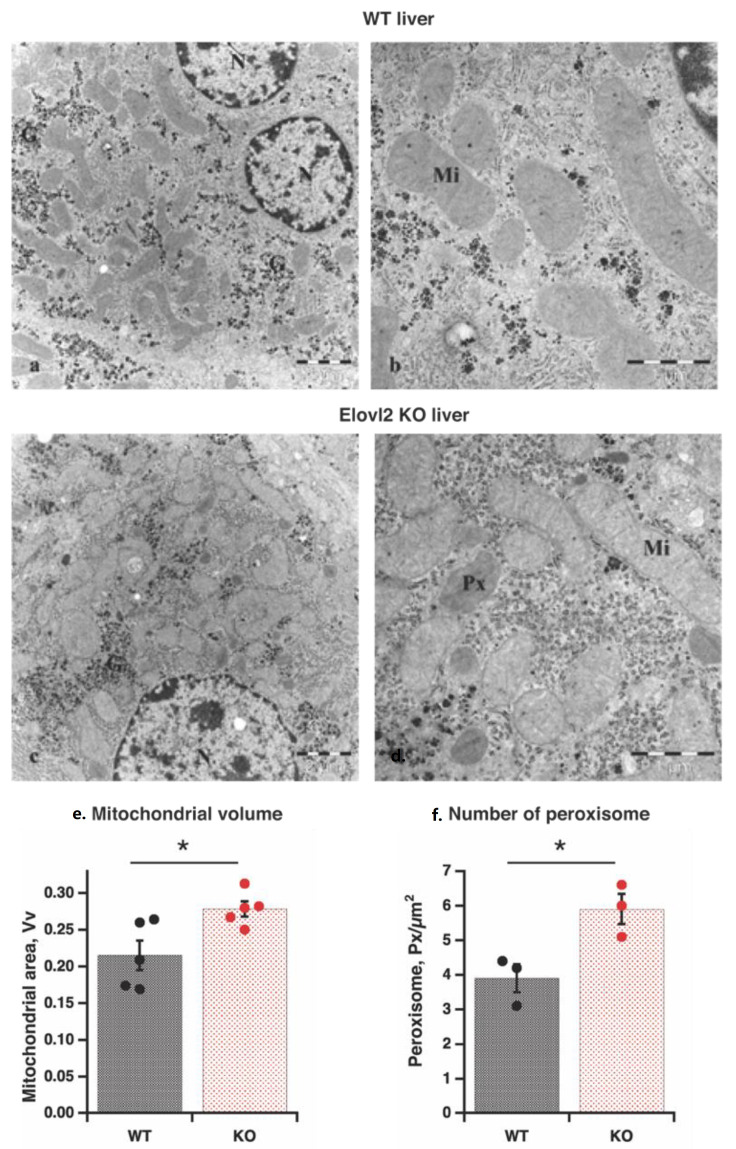
Morphology of liver tissue by electron microscopy (**a**–**d**): Transmission electron microscopy images of the liver of wild-type (WT) (**a**), and Elov2 KO (**b**) mice: mitochondria (*Mi*); peroxisomes (*Px*); nucleus (*N*); glycogen (*G*). (**e**) Mitochondrial volume. Individual values are displayed in bars with means ± standard errors of mean (SEM) from 5 different liver samples of each genotype. Significant differences are shown between WT and Elovl2 KO mice: * *p* < 0.05. (**f**) Peroxisome number. Individual values are displayed in bars with means ± SEM from three different liver samples of each genotype. Significant differences are shown between WT and Elovl2 KO mice: * *p* < 0.05.

**Figure 2 nutrients-14-00559-f002:**
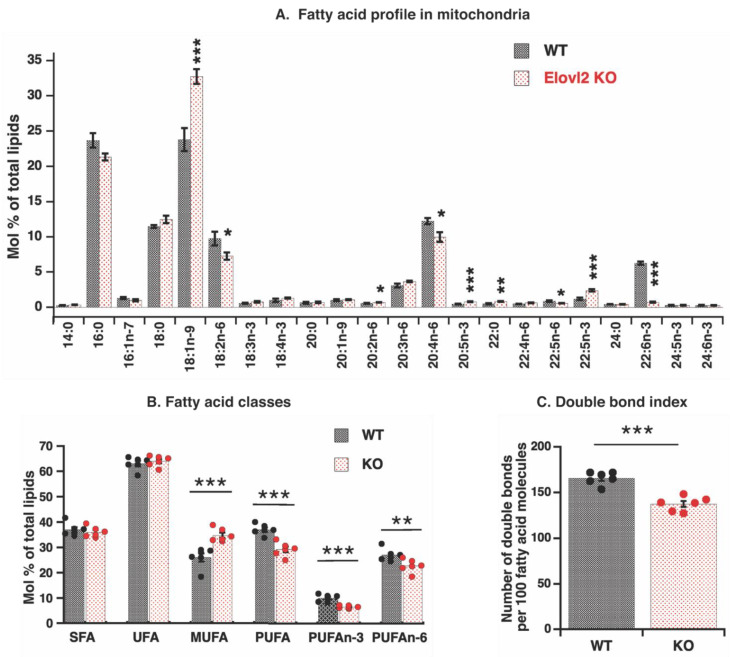
Fatty acid composition of liver mitochondria: (**A**) Membrane fatty acid composition (mol% of total mitochondrial lipids) in mitochondria isolated from the liver of WT and Elovl2 KO mice. Bars are means ± SEM from 6 different mitochondrial preparations for each mouse genotype. Values of mean, standard deviations (SD), SEM and P are shown in Appendix A. (**B**) Relative amount (mol% of total mitochondrial lipids) of different types of fatty acids: saturated fatty acids (SFA), unsaturated fatty acids (UFA), monounsaturated fatty acids (MUFA), and polyunsaturated fatty acids (PUFA). (**C**) Double bond index (DBI) in membrane fatty acids. In B and C, individual values are displayed in bars with means ± SEM from 6 different mitochondrial preparations for each mouse genotype. Significant differences are shown between WT and Elovl2 KO mice: * *p* < 0.05, ** *p* < 0.01, *** *p* < 0.001.

**Figure 3 nutrients-14-00559-f003:**
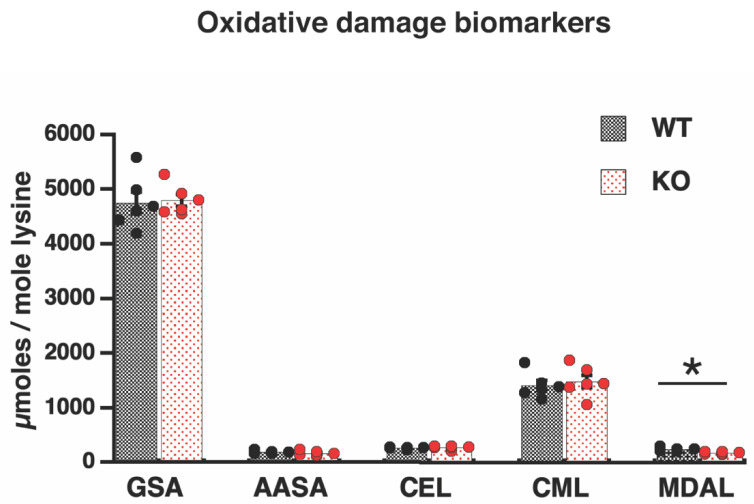
Oxidative damage markers. Lipoxidation (MDAL), protein oxidation (GSA, AASA), and glycoxidation (CEL, CML) indicators in liver mitochondria from WT and Elovl2 KO mice. Individual values are displayed in bars with means ± SEM from 6 different mitochondrial preparations for each mouse genotype. Units: µmol/mol lysine. Significant differences are shown between WT and Elovl2 KO mice, * *p* < 0.05.

**Figure 4 nutrients-14-00559-f004:**
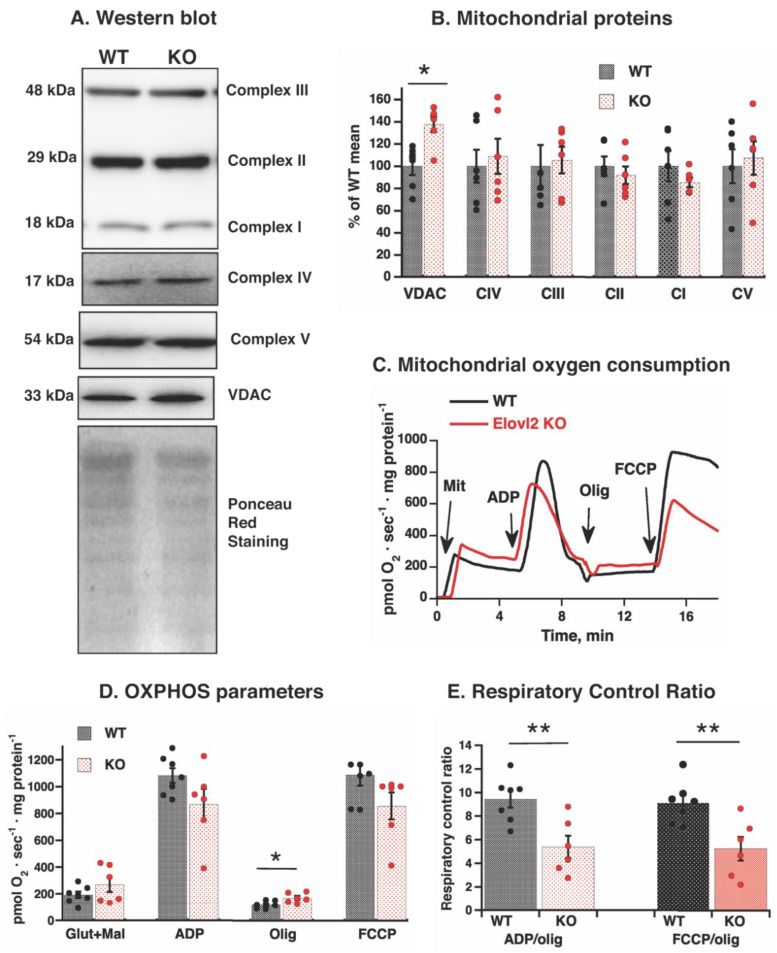
Mitochondrial respiratory function in WT and Elovl2 KO mice: (**A**) Representative western blots of structural components of oxidative phosphorylation and VDAC in liver mitochondria isolated from WT and Elovl2 KO mice. NDUFB8 subunit of Complex I, SDHB subunit of Complex II, UQCRC2 subunit of Complex III, COX4 subunit of Complex IV, ATP5A subunit of Complex V, and VDAC. In all cases, 20 µg of mitochondrial protein were loaded per lane. Ponceau red staining indicates equal loading of protein. (**B**) Quantification of western blot analysis. Mean of WT values was taken as 100%, with all other values expressed relative to this 100%. (**C**) Representative oxygen consumption traces of isolated liver mitochondria from WT mice and Elovl2 KO. Additions were 0.5 mg of liver mitochondria (*Mit*), 750 µM ADP, 3 µg/mL oligomycin (*Olig*) and 0.8 µM FCCP. (**D**) Oxygen consumption rate in response to additions of malate and glutamate, ADP, oligomycin and FCCP. (**E**) Coefficients of respiratory control ratio, RCR (ADP/oligomycin and FCCP/oligomycin) of liver mitochondria. In (**B**–**D**), individual values are displayed in bars with means ± SEM from 6 independent mitochondrial preparations of each genotype. The significant differences are shown between WT and Elovl2 KO mice, * *p* ˂ 0.05; ** *p* < 0.01.

**Figure 5 nutrients-14-00559-f005:**
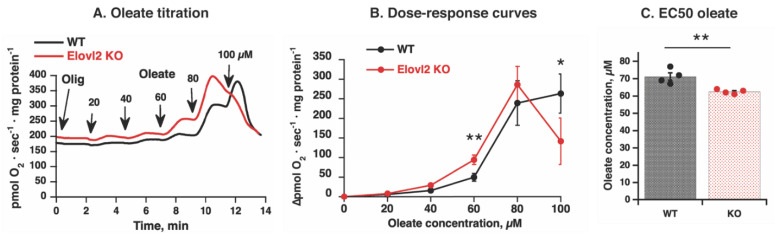
Fatty acid-induced uncoupling in Elovl2 KO mitochondria: (**A**) Representative oxygen consumption traces showing oleate effects in liver mitochondria from wild-type (WT, black line) and Elovl2 KO (KO, red line). The mitochondria were incubated with BSA and malate and glutamate, then with 3 μg/mL oligomycin (*Olig*), and titrated with oleate as indicated by arrows. Each addition of oleate was 20 µM (the final nominal concentrations of oleate are shown on the graph). (**B**) Dose-response curves for oleate were based on experiments as those shown in (**A**). The values were calculated as the increase in respiratory rate above oligomycin level. The values are the means ± SEM from 4 independent liver mitochondria isolations of each genotype performed in parallel. (**C**) Comparison between the apparent EC50 concentrations for respiratory stimulation by oleate in WT and Elovl2 KO mitochondria. Values were obtained from the dose-response curves shown in (**A**) by manual interpolations for the oleate concentration, which yielded 50% of the maximal respiratory value. Individual values are displayed in bars with the means ± SEM from 4 independent liver mitochondria isolations of each genotype performed in parallel. In (**B**,**C**), significant differences are shown between WT and Elovl2 KO mice: * *p* < 0.05, ** *p* < 0.01.

**Figure 6 nutrients-14-00559-f006:**
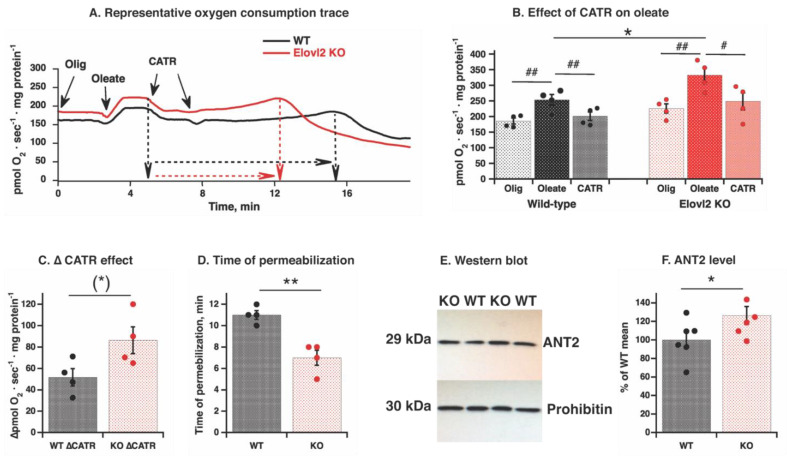
ANT2 protein level and CATR effect on fatty acid-induced uncoupling in wild-type and Elovl2 KO mitochondria: (**A**) Representative oxygen consumption traces showing effects of 60 µM oleate and 2 µM CATR in liver mitochondria. The mitochondria were incubated as in Figure 5 (**A**) with BSA and malate and glutamate, then with 3 μg/mL oligomycin (Olig), and oleate was added as a single 60 µM dose to induce detectable respiration; then treated with 1 μM carboxyatractyloside (CATR) twice as indicated by arrows. Dashed black (WT) and red (KO) arrows indicated different time durations for development of permeabilization-like phenomena. (**B**) Compilation of experiments performed as in A. The individual values are displayed in bars with means ± SEM from 4 independent mitochondrial preparations of each genotype. Significant differences are shown between WT and Elovl2 KO mice, * *p* ˂ 0.05 and between oleate-induced respiration and other respiratory states (oligomycin-insensitive and CATR-inhibited), ^#^ *p* < 0.05 and ^##^ *p* < 0.01. (**C**) CATR effect was estimated as ∆ between rate of oleate-stimulated and CATR-inhibited rate. The individual values are displayed in bars with means ± SEM from 4 independent mitochondrial preparations of each genotype. (*) indicates *p* = 0.06 (almost reaching significance between WT and Elovl2 KO mice). (**D**) Time to permeabilization after CATR addition. The time period between adding the first dose of CATR and the spontaneously developed peak of oxygen consumption was measured in minutes for each mitochondrial preparation as shown in (**A**), and the individual values are displayed in bars with means ± SEM from 4 independent mitochondrial preparations of each genotype are presented by bars. Significant differences are shown between WT and Elovl2 KO mice, ** *p* ˂ 0.01. (**E**) Representative western blot of ANT2 protein from WT and Elovl2 KO liver mitochondria. Prohibitin is used as a loading control. In all cases, 20 µg of mitochondrial protein were loaded per lane. (**F**) Quantification of western blot analysis. As a standard control, a mix of all the samples was loaded at least twice on each gel and used as a reference to calculate the amount of protein. These raw values were used for statistical analysis. For graphical presentation, normalization was performed: mean of WT values was taken as 100%, and all other values are expressed relative to this 100%. The individual values are displayed in bars with means ± SEM from 6 independent mitochondrial preparations of each genotype. Significant differences are shown between WT and Elovl2 KO mice, * *p* ˂ 0.05.

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
