# Peer review of "Elovl2-Ablation Leads to Mitochondrial Membrane Fatty Acid Remodeling and Reduced Efficiency in Mouse Liver Mitochondria"

_nutrients, 2022, doi:10.3390/nu14030559_

Round 1

Reviewer 1 Report

The manuscript of Alexia Gómez Rodrígue et al is the first work focused on understanding of the role of Elovl2 in mitochondria using the livers as a model system. Authors focus on mitochondrial structure (EM and western blots, lipid composition), function (energy metabolism and oxphos) and oxidative stress. Authors observe 1. Change in combination of lipids content in mitochondria. 2. Changed volume of mitochondria and number of peroxisomes in Elovl2KO liver cells. 3. Lack of oxidative stress increase. 4. Sensitivity of mitochondria to the fatty-acid-induced uncoupling. In general, the work is quite interesting albeit not fully developed and therefore leaving the need for few additional controls, discussion points and some reformatting data.

Suggested improvements include:

  1. Mitochondria volume is increased in ko animals. Is it because of the elongation of the mitochondria or…?
  2. What is the ratio of mitochondrial DNA/mitochondrium in wt and KO animals? Are all mitochondria functional?
  3. The genes studied by western are nuclear-coded genes, can authors check also an abundance of the mitochondria-coded proteins (one or two?). Maybe by RNA levels at least?
  4. If the amounts of mitochondrial (nucleus coded) proteins does not change and volume of the mitochondria change – what does it mean? Can authors discuss?
  5. Numbers of peroxisomes increase – however this is only checked by EM. Can authors confirm the finding by a. western blot or b. immunostaining?
  6. On the presented EM image the rough ER seem to be affected. Is it only this image, different location in the cell? Can author add lower magnification image, so it is visible? Supplementary materials could also include several images that were quantified in figure 1.
  7. Changes of ANT2 levels are quite low. Is it enough to explain the phenotype? Also, is it possible to check expression by RT-qPCR?

Technical point:

All data is presented in the “old fashion” style. It would be better to use box plots or scattered box plots to present the data. This reviewer suggests an excellent paper discussing data visualization: DOI 10.1074/jbc.RA117.000147

Minor point:

Discussing the role of Elovl2 in aging authors should cite the PMID: 31943697, the first work showing the impact of the lack of Elovl2 activity in aging-like loss of function of the organ.

Author Response

Suggested improvements include:
1. Mitochondria volume is increased in ko animals. Is it because of the elongation of the mitochondria or…?

We have now added one new TEM image in low resolution and three more TEM images in high resolution (one in updated Figure 1 and two in supplementary figure 1) of the liver for each mouse genotype. As evident from all images, mitochondrial volume is visibly higher in Elovl2 KO than WT, but the shape and structure of mitochondria are not visibly different between the two genotypes. . Therefore, we have now specified in more detail in the text (line 310) that “most morphological features (shape and structure) of mitochondria were similar in wild-type and Elovl2 KO hepatocytes”.

2. What is the ratio of mitochondrial DNA/mitochondrium in wt and KO animals? Are all mitochondria functional?

We agree with the reviewer that data about the mitochondrial DNA content could be informative. However, two of our observations, increased mitochondrial volume and sustained content of mitochondrial electron transport chain complexes in KO animals, suggested maintaining the content of mitochondrial proteins coded by mtDNA. We thus discard the need for this determination.
In our study, isolated mitochondria from the liver of wild type mice have the value 9.5 of the quality control parameter, RCR (FCCP/olig), which is considered excellent quality by scholars in the mitochondrial bioenergetics field.A low RCR value of about 3 is widely considered a threshold criterion for the quality of the isolation procedure. Typically, contamination of population mitochondria by dead mitochondria (either due to tissue pathology or the researcher’s mistakes in the procedure) is clearly reflected in the RCR by lowering it below 3.
Also, the addition of only 0.5 mg protein to the 2.0 ml incubation medium gives a very high oxygen consumption rate at the lowest sensitivity setting of an oxygraph device. Any contamination (dead mitochondria or non-mitochondrial protein) will lower oxygen consumption and require increased device sensitivity correspondingly.
Elov2 KO mitochondria exhibit lower RCR than wild-type. However, it is still higher than 5.0, excluding the presence of a significant amount of dead/unfunctional mitochondria. We cannot rule out that some mitochondria were nonfunctional. However, we can guarantee that its presence was minor and did not affect our conclusions.

3. The genes studied by western are nuclear-encoded genes. Can authors also check an abundance of the mitochondrial-encoded proteins (one or two?). Maybe by RNA levels at least?

The content of the different subunits of ETC complexes was determined by Western blot in isolated mitochondria. Analyzed subunits are core subunits meaning that they reflect the entire ETC complex content (in other words, without such a particular core subunit the whole complex will not be formed). All mitochondrial ETC complexes (except complex II) are coded by both nuclear DNA and mtDNA. mtDNA encoded subunits are also core subunits and reflect whole complex content. Thus, Western blots of subunits (whether nuclear-encoded or mitochondrial encoded) reflect the protein content of ETC complexes. This conclusion is based on our own study of the mitochondrial ETC complexes formation (PMID: 19656491 DOI: 10.1016/j.cmet.2009.06.010). The same conclusion can be made about F0F1-ATP synthase complex formation (also based on our results PMID: 17666453 DOI: 10.1096/fj.07-8581com). Wee think that an additional check of the abundance of mitochondria-encoded proteins will not provide relevant information.
Furthermore, based on our previous work with mtDNA Mutator mice (PMID: 19656491 DOI: 10.1016/j.cmet.2009.06.010)) we think that the measurement of the gene expression for ETC subunits coded from nucleus or mitochondria is not a good approach for evaluating mitochondrial protein content because the level of a given transcript does not necessarily correspond to the level of protein expression.
Consequently, we think that it is sufficient to know the mitochondrial ETC content with the approach used in our current study.

4. If the amount of mitochondrial (nucleus coded) proteins does not change and volume of the mitochondria changes – what does it mean? Can authors discuss?

We have to emphasize that the content of mitochondrial proteins has been measured in isolated mitochondria here, in the same mitochondria samples used for functional analysis and lipid analysis. Three components (protein, lipid, and oxidative efficiency) were analyzed in the same mitochondrial population, and it was concluded that the lipid component is responsible for changes in oxidative efficiency (not ETC proteins alone). We believe that our results are sufficiently discussed.
However, we can guess that the reviewer is interested in the mitochondrial protein content in whole tissue versus mitochondrial volume. First - we do not have measurements of mitochondrial proteins in whole tissue, and therefore we cannot discuss absent results in our manuscript. However, we can discuss a little bit here: according to a general belief in the mitochondrial field, the ETC protein measured in whole tissue gives only an approximation of total mitochondrial volume/density. The best and most accessible estimation of mitochondrial volume/number is TEM (used here). Note! There is, of course, more advanced microscopy than TEM used here. Other molecular or enzymatic mitochondrial markers (citrate synthase activity, mtDNA amount, cardiolipin content) are often used. However, these markers can be affected by nutritional status, exercise, pathology, etc., and must be considered with precautions.

5. Numbers of peroxisomes increase – however this is only checked by EM. Can authors confirm the finding by a. western blot or b. immunostaining?

We agree with the reviewer that this finding on the increased number of peroxisomes needs to be studied in detail further, and we are currently planning extensive studies on adaptive changes in peroxisomal fatty acid elongation pathways. In addition, peroxisomal enzyme activities will be measured and peroxisomal contribution to ROS production/protection. The current work is devoted to mitochondrial studies, and peroxisomes are just briefly mentioned here since TEM is shown. To put additional emphasis on peroxisomes will break the scope of the current story. However, we have now described the future perspectives of peroxisomal studies in the discussion, adding “Another interesting future perspective is related to the involvement of peroxisomal pathways and desaturases (Fads1 and Fads2) in adaptive modification of the fatty acid composition of mitochondrial membranes” (lines 7672-675).

6. On the presented EM image the rough ER seems to be affected. Is it only this image, a different location in the cell? Can the authors add a lower magnification image, so it is visible? Supplementary materials could also include several images that were quantified in figure 1.

As the reviewer requested, we have now added one new TEM image at low magnification and three TEM images at high magnification (one in updated Figure 1 and two in supplementary figure 1) of the liver for each mouse genotype. As evident from all images, the rough ER looks slightly different in only one wild-type image (this image is now placed in the supplement). The wild-type image in main text figure 1 is substituted by a new image, which is considered more representative and chosen by the recommendation of an EM specialist. We have now specified in more detail in the text (lines 317-319) that “there are no visible differences in appearance of ER, nucleus and content of glycogen between wild-type and Elovl2 KO livers.”

7. Changes of ANT2 levels are quite low. Is it enough to explain the phenotype? Also, is it possible to check expressions by RT-qPCR?

ANT2 alone is not enough to explain the phenotype. Moreover, this carrier could be involved in regulating inducible proton leak only under conditions of high endogenous non-esterified fatty acids. Therefore, we indicate in the discussion, “Further research to identify additional pathologic or physiological in vivo conditions with elevated levels of non-esterified fatty acids is required to confirm the significance of ANT involvement in the mechanism of metabolic inefficiency.”
However, as the reviewer requested, RT-qPCR was performed, and the description of the Real-Time qPCR was added to the Methods section (lines Methods 221-238). The results demonstrated that ANT2 gene expression was significantly increased in KO mice compared to WT (Supplementary Figure 4). In addition to the Western Blot data, this result could confirm that the ANT2 amount in KO animals is upregulated.
The following text was added to the results section: ”To confirm higher protein expression, Ant2 gene expression measurement in whole liver tissue has been performed by Real-Time qPCR (Supplementary Figure 4). Ant2 gene expression was 48% higher in the liver from Elovl2 KO than from wild-type. Higher Ant2 gene expression may reflect both higher ANT2 protein within the mitochondrial population as well as higher total mitochondrial synthesis (evident from higher tissue mitochondrial volume, Figure 1E). Western blot and Real-Time qPCR of ANT1 was also performed, but no clear band was detected on Western blot and very high Ct values in Real-Time qPCR (not shown), confirming that ANT2 is exclusively expressed in mouse liver”. (lines 550-558)

Technical point:
All data is presented in the “old fashion” style. It would be better to use box plots or scattered box plots to present the data. This reviewer suggests an excellent paper discussing data visualization: DOI 10.1074/jbc.RA117.000147 https://www.sciencedirect.com/science/article/pii/S0021925820327927?via%3Dihub

We thank the reviewer for this suggestion. As the reviewer suggested, we have changed six “old fashion” graphs to “modern” dot-plots. However, some graphs (Figure 2, Figure 3, and 4) included several bars we decided to keep in the original format because in dot plots, they look too crowded, and it is difficult to see differences in small amounts of fatty acids, for example.

Minor point:
Discussing the role of Elovl2 in aging authors should cite the PMID: 31943697 the first work showing the impact of the lack of Elovl2 activity in aging-like loss of function of the organ.

As the reviewer suggested we have added the following paper that showed the impact of the lack of Elovl2 activity in retinal aging in the discussion (line 613-615) and in the references list.
40. Chen, D.; Chao, D.L.; Rocha, L.; Kolar, M.; Nguyen Huu, V.A.; Krawczyk, M.; Dasyani, M.; Wang, T.; Jafari, M.; Jabari, M., et al. The lipid elongation enzyme elovl2 is a molecular regulator of aging in the retina. Aging Cell 2020, 19, e13100.

Reviewer 2 Report

The manuscript provides the effect of Elovl2 ablation on mitochondrial function and fatty acid remodeling , bioenergetics and oxidative stress markers. The research design is good.

But there are some points to be modified.
1. line 54, the full name DHA and DPA  should be given. For other abbreviations, they should be given the full name for the first time too.

2. line 74, what have you found from this research should be moved to the discussion section rather than introduction section.  And in results section, literatures may be moved to the discussion too, it is better to display your own results in the results section.

3. line 269 and 273 , DHA were significantly lower in Elovl2 KO mice, and 22:5 n-3 were significantly increased, and 20:5n-3 increased too shown from the Figure 2, but for 18:3 n-3, precursor of DHA, there was no difference between two groups. what is the reason? authors compared the relative amount of fatty acid levels(%), if the absolute concentration of 18:3 n-3 was used , there may be different findings. 

Author Response

The manuscript provides the effect of Elovl2 ablation on mitochondrial function and fatty acid remodeling, bioenergetics and oxidative stress markers. The research design is good.

But there are some points to be modified.

1. line 54, the full name DHA and DPA should be given. For other abbreviations, they should be given the full name for the first time too.

Thank you for noticing. It is corrected now. Moreover, all other abbreviations were checked.

2. line 74, what have you found from this research should be moved to the discussion section rather than the introduction section.  And in the results section, literature may be moved to the discussion too, it is better to display your own results in the results section.

From Nutrients “Instructions for Authors” https://www.mdpi.com/journal/nutrients/instructions “Introduction: The introduction should briefly place the study in a broad context and highlight why it is important. It should define the purpose of the work and its significance, including specific hypotheses being tested. The current state of the research field should be reviewed carefully, and key publications cited. Please highlight controversial and diverging hypotheses when necessary. Finally, briefly mention the main aim of the work and highlight the main conclusions.”

We are following these instructions and reviewing “the current state of the research field,” revealing the gap in the literature with “However, little is known.. and defining the purpose of the work. Therefore, we think this part from line 74 is important for the introduction. However, we considered the reviewer’s point and added a few words, such as “It is known that” and “In addition, previous studies” to specify that this literature is from prior research essential for our problematization and aim.  

There are only a few references on literature in the results section, which are necessary for a logical flow of explanation/interpretation of the results and connect the results subsections. Here we are following the Nutrients “Instructions for Authors” Results: Provide a concise and precise description of the experimental results, their interpretation as well as the experimental conclusions that can be drawn.”

Without these few references, readers may be lost in understanding the logic of the presentation of the results. However, we agree that primary literature is essential in the discussion, where we have it extensively.

3a. Line 269 and 273, DHA were significantly lower in Elovl2 KO mice, and 22:5 n-3 were significantly increased, and 20:5n-3 increased too shown from the Figure 2, but for 18:3 n-3, precursor of DHA, there was no difference between two groups. What is the reason?

As the reviewer indicates, DHA was significantly lower in Elovl2 KO mice, whereas both 20:5 n-3 and 22:5n3 were significantly increased. By contrast, for 18:3 n-3, a precursor of DHA, there was no difference between the two groups. As stated in the text, this observation clearly indicates the steps in which Elovl2 and peroxisomal oxidation are working in omega 3 (n-3) fatty acid biosynthesis. These findings, along with changes in fatty acids from PUFAn-6 pathway, where Elovl2 also participates, and the increase in 18:1n9 content seems to suggest an adaptive response in KO animals in order to maintain stable the SFA and UFA contents and, in turn, the biophysical properties of the mitochondrial membrane within physiological limits. This observation is in line with the idea that cell membranes have allostatic adaptive response mechanisms designed to preserve their composition and function within homeostatic limits (Hagen RM, Rodriguez-Cuenca S, Vidal-Puig A. An allostatic control of membrane lipid composition by SREBP1. FEBS Lett. 2010; 584(12):2689-2698).

The sustained content of 18:3n3, as essential fatty acid participating in mitochondrial lipid composition, but with a minor content in comparison with the other fatty acids, may be interpreted in terms that the supply of this fatty acid is not altered in KO animals and participates in membrane lipid composition in a usual way. As expected, changes are observed in the biosynthesis pathway, not in the content of this precursor.

In case the reviewer’s thought was that since DHA is decreased, the precursor must be increased because biosynthesis of PUFAn3 is reduced, we can say that values are: 0.55±0.06% 18:3n3 in the control group vs. 0.73±0.11% in KO animals. Thus, it seems that there is an apparent increase (30%) in KO animals, although this increase does not reach a significant difference (p=0.212).

Furthermore, we can add for discussion that our previous study, “Effect of maternal and offspring genotype and maternal diet on offspring hepatic gene expression” (Pauter AM et al. 2016; DOI 10.1194/jlr.M070862), showed that the expression of Fads1 and Fads2 showed a tendency to be higher, especially in the Elovl2−/− pups fed by Elovl2−/− mothers maintained on the control diet. It is known that FADS2 is responsible for the desaturation of 18:3n3 to 18:4n3. Thus, upregulation of this desaturase could be evident in Elovl2 KO in our study. We plan to measure such adaptive changes in desaturases parallel to peroxisomal upregulation (as discussed in response to Reviewer 1). Added text “Another interesting future perspective is related to the involvement of peroxisomal pathways and desaturases (Fads1 and Fads2) in adaptive modification of the fatty acid composition of mitochondrial membranes” (lines 672-675). 

3b. Authors compared the relative amount of fatty acid levels (%), if the absolute concentration of 18:3 n-3 was used, there may be different findings. 

We agree with the reviewer that there might be different findings if the absolute concentration of 18:3 n-3 was used. However, no differences in the contents of major mitochondrial proteins were observed, as well as no significant visible differences in mitochondrial structures was revealed by TEM. Consequently, there is no reason to think of a significantly different lipid bulk content. In addition, from a methodological point of view, an amount equivalent to 200 micrograms of mitochondrial proteins were used for all samples to analyze the fatty acid profile. Furthermore, during chromatogram analysis, no differences in the integration area for the different peaks suggestive of differences in the absolute amount of lipids were observed between samples. Thus, we think that expressing the relative amount of fatty acid levels as % is a practical approach to know the mitochondrial membrane fatty acid composition.

Round 2

Reviewer 1 Report

This reviewer is satisfied with the introduced changes. 

One important edit should be introduced, though. Since authors do not want to present all pieces of data in dot plots (with or without bars) it should be noted on each panel how many replicates of given experiment was done. We believe the oxidative stress markers, double bond index and quantification in respiratory chain can still be presented better as scattered plots (with standard dev). 

Author Response

All figures (except Figure 2A) are now presented as scattered individual values in bars with Means ± Standard mean errors (SEM). Moreover, Figure 2A is additionally shown as Supplementary Table with standard deviation (SD), N, and P.